# Normative reference values of handgrip strength for Brazilian older people aged 65 to 90 years: Evidence from the multicenter Fibra-BR study

**Michael Eduardo Reichenheim**[1☯¤a], **Roberto Alves Lourenço**[2,3☯], **Janaína Santos Nascimento**[3,4☯], **Virgílio Garcia Moreira**[3☯]*, **Anita Liberalesso Neri**[5‡], **Rodrigo Martins Ribeiro**[3‡], **Lygia Paccini Lustosa**[6†‡], **Eduardo Ferriolli**[7‡]

1 Institute of Social Medicine, Rio de Janeiro State University, Rio de Janeiro, Brazil, 2 Department of Medicine, Pontifical Catholic University, Rio de Janeiro, Brazil, 3 Research Laboratory on Human Aging—GeronLab, Internal Medicine Department, Faculty of Medical Sciences, State University of Rio de Janeiro, Rio de Janeiro, Brazil, 4 Occupational Therapy Department, Federal University of Rio de Janeiro, Rio de Janeiro, Brazil, 5 Department of Educational Psychology, Faculty of Education, State University of Campinas, São Paulo, Brazil, 6 School of Physical Education, Physiotherapy and Occupational Therapy, Federal University of Minas Gerais, Belo Horizonte, Brazil, 7 Department of Internal Medicine, Ribeirao Preto Medical School, University of São Paulo, São Paulo, Brazil

☯ These authors contributed equally to this work.
† Deceased.
¤a Current address: Institute of Social Medicine, Rio de Janeiro State University, Rio de Janeiro, Brazil
‡ These authors contributed equally to this work.
* virgilio.garcia.moreira@gmail.com

## Abstract

### Background

Handgrip strength (HGS) is an indicator of muscle strength, suited for evaluating the aging process. Its use depends on the availability of reliable normative reference values (NRV). The main objective of this study is to provide NRV of HGS for Brazilians aged 65 to 90 years.

### Methods

Participants were from the Frailty in Brazilian Older People research. 2,999 successful aging (SA) participants comprised the development sample. HGS was measured using a hydraulic dynamometer. Obtaining NRV involved regressing HGS on age per sex-height strata, fitting separate fractional polynomial (FP) models for the mean and coefficient of variation. Model fit was assessed via standardized residuals, probability/quantile plots, and comparing observed to normal expected percentages of participants falling within specified centile intervals. For validation, the latter procedure was applied to 2,369 unsuccessfully aging (UA) participants.

### Results

Across strata, the best-fitting models for the means were FP of power 1. FP models for the CV indicated age invariance, entailing steady heteroscedastic age decline in SD since coefficients for the means were negative and SD = CV×mean. All models adjusted well. Centiles

**Data Availability Statement:** All relevant data are within the manuscript and its Supporting Information files.

**Funding:** This work was supported by the Brazilian National Research Council (CNPq) cnpq.br [555087/2006-9 – RAL] and the Carlos Chagas Foundation for Research Support (faperj.br), State of Rio de Janeiro [171.469/2006 – RAL]. M.E.R. was partially supported by the CNPq (grant number 301381/2017-8). The funding sources had no role in the designing and conduction of the study; collection, management, analysis, and interpretation of the data; and preparation, review or approval of the manuscript.

**Competing interests:** We explicitly declare that there are no conflicts of interest.

distributions for the SA and UA populations showed anticipated patterns, respectively falling on and below the normative expected centile references. Results (NRV) are presented in tables and centile charts. Equations are also provided.

## Conclusion

NRV/charts may be endorsed for routine use, while still tested further. They would aid professionals caring for older people, not only to identify those at risk and eligible for immediate provisions, but also in planning prevention and rehabilitation measures.

## Introduction

The aging of the skeletal muscle system is a complex process involving loss of muscle mass and function, reduction of bone mass, degenerative changes in the tendons, ligaments, intervertebral discs, and articular cartilage. Studies have shown that between the second and fourth decade of life, muscle mass and strength reach their maximum and begin to decrease thereafter. Muscle strength is an expression of this senescence, thus serving as a potential biological marker of physiological aging [1].

Regarded a good indicator of muscle strength, handgrip strength (HGS) has been proposed as a tool for use in the evaluation of the aging process [2–5]. The indicator is equivalent to other measures such as respiratory, trunk and lower limb muscle strength, but offers the advantage of being easy to use in clinical practice due to its low cost, good portability, and not requiring specialized equipment and personnel [6, 7].

HGS has been widely used in health services for older people in the last three decades. In practice, the indicator has been employed as a predictor of multiple negative outcomes befalling old age and in the assessment of the effectiveness of interventions and ensuing prognosis [8, 9].

HGS has also been utilized to demarcate risk situations, assisting health professionals in their decision-making processes. As part of the screening process, for example, HGS was included as an item on a frailty scale [8] and in a proposed algorithm for selecting individuals for later muscle mass assessment [10]. In fact, there is a current debate on how and in what stage of a serial test sequence should HGS come in [11]. Beyond the contexts of individual care, HGS has also been adopted to delimit strata in empirical studies, both in clinical and epidemiological research focusing on population surveillance or causal explanations.

Regardless of its intended application, HGS has commonly been employed as a stand-alone measure, given some agreed cut-off point to demarcate a risk situation. Several studies have sought to provide empirical referencing of the HGS. Regarding older people, most recommendations involve just a few and all-encompassing cut-off points [2, 4, 5]. One obstacle to this approach is that it ignores ethnic diversity [12]. Besides, even within a certain population domain, HGS differential concerning age, sex and anthropometric characteristics (e.g., height) are well documented [13]. Demarcations that respect differences would be a step forward in both clinical practice and research.

There is also a prospective perspective, in which the muscle strength indicator is evaluated sequentially in time. In the context of health care for older people, this approach envisages monitoring the evolution of those assigned regularly to health services. The indicator may be useful to flag health decline, aiding professionals in identifying that something is out of order and requiring further propaedeutic investigation (e.g., Comprehensive Geriatric Assessment)

[14]. This perspective is most valuable in overseeing older people undergoing therapeutic or rehabilitation interventions [15].

The establishment of HGS as an indicator of muscle mass and strength loss involves a two-step process: classifying subjects according to a standard 'normal' population and identifying and agreeing upon demarcation points to be employed in decision-making processes. The first step is crucial in sustaining the second. Participants must be first adequately rated vis-à-vis an expected benchmark defined by a population recognizably aging healthily. Only when this step reaches a certain level of development may one look for inflection points that allow the proper delimitation of increasing levels of risk and decline. Identifying HGS centile thresholds indicative of impending falls, cognitive impairment, or even death, only makes sense if participants are properly ranked first.

Although the related literature sometimes conflates both steps, a scrutiny of the studies responding to the first step–classification–roughly reveals two levels of detailing. In one camp are studies proposing normative reference values (NRV) of HGS according to few and fixed centile groups, usually stratified by sex and some coarse age stratification [2, 4, 5, 16, 17]. Some further disaggregated NRV by height [3, 17]. The other camp involves studies presenting NRV along with comprehensive reference charts [18–22] or, alternatively, providing equations for projecting refined percentile groups of HGS according to several predictors such as age, sex, anthropometric measures or hand domination whereby centile charts may be produced [23]. Of note is that graphs are more informative by design since age is implicitly discretized regardless of the other levels of stratification. Although often physiological, the prominent decline in HGS after 65 years of age makes this feature particularly important.

Albeit in progress and off to a favorable start, this literature has some drawbacks that require addressing. As mentioned, a major downside regarding the first type of studies (camp) is the inability to age refine classifications since studies are limited to few discrete strata (sex and 1–5 age groups) [2, 4, 5, 16, 17]. Moreover, all but two studies [3, 17] lack further stratification beyond sex. Besides hindering cross-sectional classifications in one-off assessments, this lack of discretization is a real obstacle to the goal of following up patients or participants under study.

While studies of the second camp tend to avoid these drawbacks, their chosen reference populations require judgment. Three out of the six reviewed studies did not perform any sub-selection to remove individuals with explicit health problems or showing physical or cognitive impairments [18, 21, 22]. Although the other three studies did apply some eligibility criteria [19, 20, 23], their restriction as to the reference healthy population was confined to a few conditions unrelated to cognition and functionality. These studies hardly meet the lately proposed concept of successful aging (SA) that brings together essential qualifiers [2, 4, 5, 16–21]. Another weakness follows from this lack of rigor, since, complementarily, there are also no validation studies using external populations overtly experiencing unsuccessful aging (UA). These issues require redressing.

The state of the art concerning the Brazilian literature on the subject also matters to this study. As conveyed in earlier references, there are only two studies so far, both published almost 10 years apart [2, 16]. Of importance, neither report or propose reference charts for practical use in health care or research settings. Besides, they are circumscribed geographically thus impairing their findings to be extended to the country at large.

In sum, despite the recognized standing of HGS in assessing risk of older populations, its adequate use depends on the availability of reliable NRV to distinguish weaker from more apt individuals [10, 24–26]. In fairness, the literature does show studies covering classification and risk demarcation but is rather empty in regard to defining NRV based on a (study) population explicitly undergoing a SA course. Moreover, as the reviewed literature shows, the NRV

proposed so far present considerable disparities due to ethnic, phenotypic or even cultural variations [27, 28]. To meet the expectations placed on the HGS indicator for years, there are thus still gaps in the knowledge regarding the proper classifications of older people that also account for regional and/or cultural differences and particularities. Therefore, the main objective of this study is to provide NRV (and respective charts) of HGS for Brazilians aged 65 to 90 years, stratified by sex and height strata, based on a reference population known to be aging healthily. An ancillary objective is to validate the proposed centile based NRV (and respective charts) using an external population regarded as aging unsuccessfully.

## Materials and methods

### The FIBRA-BR study

The present study comprised individuals from the Frailty in Brazilian Older People research network database (Fibra-BR). The methodological description of the Fibra-BR study has been provided elsewhere [29]. Briefly, this is a multicenter, population-based, cross-sectional study conducted between January 2009 and January 2010. The primary aim was to investigate the prevalence of frailty and associated factors in Brazilian older people.

Previously trained field teams interviewed and examined 65 years old and older individuals from 16 cities covering regions of Brazil of different sociodemographic characteristics. Cities and source populations were selected per operational convenience, but a probabilistic sex-age-stratified sampling procedure was performed at each site. Older people with physical or sensory conditions precluding the assessment of HGS were excluded. Thus 7,447 older people were eligible for further selection (as described below).

The study followed the guidelines of the Brazilian National Committee for Ethics in Research of the Health Ministry for Research with Human Beings. All participants signed an informed consent form, and the Research Ethics Committees of the State University of Campinas, the University of Sao Paulo - Ribeirao Preto, the Federal University of Juiz de Fora and the Pedro Ernesto University Hospital of the State University of Rio de Janeiro (numbers 208, 5018, 313 and 1850, respectively) approved the research protocol.

### Defining the reference and validation populations

The present study defined two subsamples. One consisted of participants presumed ageing successfully and used to develop the NRV (henceforth referred to as reference sample). SA is a concept tapping into the physical, cognitive, emotional or social aspects related to the aging process, and has been demarcated through several facets [30]. Seven of those have been most frequently considered: 1. cognitive performance; 2. social support; 3. self-perception of health; 4. physical performance; 5. social participation; 6. independence in daily life activities; and 7. satisfaction with life. The SA construct used here was adapted from the models proposed by Rowe & Kahn and Canedo, Lopes & Lourenço [31, 32], as well as those presented in the systematic review carried out by Cosco et al. [30].

We defined the reference population as necessarily endorsing SA criteria 1, 4 and 6. Participants were considered cognitively preserved (criterion 1) if scoring above 18 points (illiterate participants) or above 23 points ($\geq$ 1 year of schooling) in the Mini-Mental State Examination (MMSE) [33]. Gait speed scoring above 0.8 m/s over 4.6 meters defined adequate physical performance (criterion 4). Individuals were considered independent (criterion 6) in the absence of impairment in basic activities of daily living (ADL) evaluated by the Katz Index [34]. The second subsample was specified for validation purposes (henceforth referred to as validation sample) and comprised older people with low cognitive performance. This sample consisted thus of exclusively older people with flagrantly aging unsuccessfully.

Prior to dealing with outliers (as described in the next section), the reference and validation samples consisted of 3,128 and 2,447 individuals, respectively. All statistical analyses stratified HGS by sex and height (in meters) according to the following cut-off points: males, >1.7, >1.6–1.7 and ≤1.6; females, >1.6, >1.5–1.6 and ≤1.5. This choice of cutoffs was guided by background knowledge and backed up by interim exploratory assessments.

## Measures and measurements

A multi-thematic questionnaire was applied to all invited participants [29], covering sociodemographic, health/cognitive and anthropometric variables. Of interest here were sex, age, height and HGS. Age was calculated from subtracting the registered date of the interview from the reported date of birth. Height was measured with a graduation rule fixed on a wall, with the participants instructed to stand up barefoot in the erect position looking at the horizon with their backs to the wall.

HGS was measured using a JAMAR hydraulic dynamometer, Model J00105, manufactured by Lafayette Instruments (3700 Sagamore Parkway North, PO Box 5729 Lafayette, IN 47904 USA). For the test, participants were seated in chairs without arm support, and feet flat on the floor. The examiner adjusted the dynamometer in the dominant hand of the individual, who had to remain with the shoulder attached to the chair, elbow flexed at 90˚, forearm in a neutral position (thumb up), and the wrist in a comfortable position, according to the American Society of Hand Therapists. The procedure was repeated three times.

Following literature [35] and a preliminary scrutiny of measurements, we chose to use only the mean value of the second and third assessments, once observing that these were much closer to each other and markedly different from the first. The analyses excluded 273 participants with biologically implausible mean values, i.e., males with HGS ≥ 65 and females with HGS ≥ 45 [36]. So too were 252 with a difference above 5 kgf between the second and third measurements. This cut-off followed an inspection of the Bland & Altman 95% agreement limits of [−5,325; 5,680] for men and [−4,613; 4,350] for women [37]. An overall, approximate value of ± 5 kgf was chosen for simplicity. Another six outlying individuals were excluded from the reference sample to ensure model adequacy (Gaussian distribution of the standardized residuals, as described in the next section). Applying all exclusions, the effective samples used in the reference development and validation stages amounted to 2,999 and 2,369 participants, respectively.

The following additional variables were used to describe the study population profile: schooling (illiterate, 1–4 years, five years or more), race (white and other than white), body mass index / BMI, number of reported morbidities (0; 1–2; ≥ 3) and independent and dependent instrumental activities of daily living / IADL [38].

## Statistical analysis

**Development of age-specific centile values and reference intervals.** As conveyed before, this stage used the reference sample. Three issues were addressed before proceeding to the main analyses. We first assessed whether the analyses needed accounting for a city clustering effect, fitting a random intercept model to the HGS variable [39]. Next, we evaluated the normality assumption of HGS and the need for any transformation otherwise [40]. Third, heteroscedasticity was examined to guide how to further specify the main models, especially regarding modeling the standard deviate (SD) curves and thence projecting the age-related reference centiles. At issue is whether one may assume an age-constant SD for HGS; and, if not, to proceed with an alternative procedure, such as modeling the coefficient of variance (CV) instead (as explained further on). We employed the Breusch-Pagan [41] and Cook Weisberg [42] tests for heteroscedasticity.

From several statistical methods available for constructing age-related NRV and centile charts [43–45], we adopted the approach recommended in Altman and Royston (1993) apud Altman & Chitty (1993) and Royston & Sauerbrei (2008) [46, 47]. By strata (sex and height), we first searched for the best fitting fractional polynomial (FP) model for the mean and SD. Since heteroscedasticity was rejected in four out of six strata (see Results), we thus opted to deal with this issue by modeling the dispersion using the coefficient of variation (CV) instead [48]. Respecting the preliminary scrutiny of the data, all models were linear regressions of HGS on age polynomials. The procedure used Stata's *xrigls* program presented in Wright & Royston (1998) [49, 50]. These best fitting models yielded the needed equations for calculating age specific percentiles and the model based smoothed HGS.

Model adequacy was first assessed by investigating the distribution of standardized residuals–z scores [47]. The z-scores are expected to have an age-independent standard normal distribution if the model is correct. This appraisal was aided by inspecting the scattergrams of these z-scores by age (per sex-height strata), along with the standardized normal probability plots (pnorm) and the quantiles of z-score against quantiles of normal distribution plots (qnorm). Finally, we compared the observed percentages of participants falling within the –1 and +1 SD and –2 and +2 SD intervals with the normal (gaussian) expected percentages (68.3% and 95.5%, respectively).

Regression coefficients and ensuing equations arising from the uncovered best fitting FPs allowed calculating the predicted HGS values and obtaining the corresponding reference charts. The equations also permitted calculating centile values for all the older people, including those comprising the validation sample (see next section). The equations per se are provided in the results section, once the uncovered 'best' FP are shown.

**Validation procedure.** We replicated the comparison between the observed and expected percentages of individuals falling between –1 and +1 SD and –2 and +2 SD, now using the validation sample comprised by older people with UA. We anticipated a much lower than expected percentage within these bounds, and a larger number of individuals outside the limits. An ancillary assessment sought to contrast the observed and expected percentages vis-à-viz specific centiles, in particular at how many older individuals would fall above the upper (80th, 90th, 97th) or below the lower (20th, 10th, 3rd) centiles than expected.

## Results

The study population profile is shown in Table 1. Most of the sample comprised females and participants less than 75 years old. Compared to the reference sample, the validation sample involved individuals with low educational level and dependent for the IADLs.

The random intercept model fitted to the aggregate reference sample to examine a possible clustering effect showed a between-city variance of 6.795 (95% CI: 3.211; 14.381) and a within-city residual variance of 79.872 (95% CI: 75.92; 84.031). This almost 12-fold lower variance indicated that most of the variation occurred at the individual level rather than across cites. However, the Intraclass Correlation Coefficients still showed substantial amounts of within-cluster HGS variation in all strata [40] a) male, >1.7: 0.172; (b) male, >1.6–1.7: 0.094; (c) male, ≤1.6: 0.104; (d) females, >1.6: 0.151; (e) female, >1.5–1.6: 0.124; and (f) female, ≤1.5: 0.1709. Table 2 conveys the city-specific HGS mean values and respective SD, as well as the distances from the grand mean (25.3 kgf) and standard deviations (9.3). For visual inspection, see S1 Fig.

These clustering effects were regarded as non-trivial and, therefore, in need of addressing in the modeling process. To this end, all ensuing analyzes used the generalized Huber-White-sandwich robust estimator to relax the assumption of independence. Note, however,

**Table 1. Characteristics of the study, reference and validation populations regarding sociodemographic, clinical and functional aspects.**

| Strata | | Total sample | | Reference sample | | Validation sample | |
|---|---|---|---|---|---|---|---|
| | | N | (%) | N | (%) | N | (%) |
| **Sex** | | | | | | | |
| | Male | 2449 | (32.9) | 1280 | (40.9) | 668 | (27.3) |
| | Female | 4998 | (67.1) | 1848 | (59.1) | 1779 | (72.7) |
| **Age (years)** | | | | | | | |
| | 65–74 | 4589 | (61.7) | 2219 | (70.9) | 1331 | (54.4) |
| | 75–84 | 2409 | (32.3) | 837 | (26.8) | 875 | (35.8) |
| | ≥ 85 | 449 | (6.0) | 72 | (2.3) | 241 | (9.8) |
| **Race** | | | | | | | |
| | White | 3878 | (52.4) | 1792 | (57.6) | 1166 | (48.0) |
| | All others | 3524 | (47.6) | 1321 | (42.4) | 1261 | (52.0) |
| **Schooling (years)** | | | | | | | |
| | 0 (illiterate) | 1297 | (17.4) | 634 | (20.3) | 258 | (10.5) |
| | 1–4 | 3258 | (43.8) | 1044 | (33.4) | 1551 | (63.4) |
| | ≥5 | 2892 | (38.8) | 1450 | (46.3) | 638 | (26.1) |
| **Physical morbidities**[a] | | | | | | | |
| | 0 | 1238 | (18.0) | 590 | (18.9) | 417 | (17.3) |
| | 1–2 | 3552 | (51.6) | 1696 | (54.2) | 1227 | (51.0) |
| | ≥3 | 2093 | (30.4) | 840 | (26.9) | 760 | (31.6) |
| **IADL** | | | | | | | |
| | Independent | 3603 | (48.4) | 1967 | (62.9) | 1054 | (43.1) |
| | Dependent | 3844 | (51.6) | 1161 | (37.1) | 1393 | (56.9) |
| **BMI** | | | | | | | |
| | Underweight | 1001 | (13.6) | 384 | (12.4) | 381 | (15.7) |
| | Normal weight | 2957 | (40.0) | 1303 | (41.9) | 923 | (38.2) |
| | Overweight | 1655 | (22.4) | 754 | (24.2) | 487 | (20.1) |
| | Obese | 1772 | (24.0) | 669 | (21.5) | 628 | (26.0) |
| MMSE ($\mu \pm$ SD) | | 24.18 | (±3.42) | 26.22 | (±2.62) | 21.64 | (±2.35) |
| HGS 2 ($\mu \pm$ SD)[b] | | 22.89 | (±9.55) | 25.65 | (±9.87) | 20.88 | (±9.02) |
| HGS 3 ($\mu \pm$ SD)[c] | | 22.94 | (±9.41) | 25.62 | (±9.68) | 20.92 | (±8.90) |
| Height ($\mu \pm$ SD) | | 1.57 | (±0.94) | 1.59 | (±0.09) | 1.55 | (±0.09) |

$\mu$: Mean; SD: standard deviation.

[a] Variable with missing data.

[b] Second assessment of handgrip strength (kgf).

[c] Third handgrip strength assessment (kgf).

that this had no direct influence on the development of the age-specific reference intervals since projected HGS estimates and related centiles depend exclusively on the point-estimates of coefficients for the mean and dispersion models (standard deviate or coefficient of variation).

The normality assumption of HGS was sustainable in all sex-height strata. The skewness / kurtosis tests showed the following p-value: (a) male, >1.7: 0.9216 / 0.3230; (b) male, >1.6–1.7: 0.5118 / 0.0169; (c) male, ≤1.6: 0.4709 / 0.2734; (d) females, >1.6: 0.9073 / 0.0185; (e) female, >1.5–1.6: 0.5711 / 0.0001; and (f) female, ≤1.5: 0.0591 / 0.0000. Although the absence of kurtosis could not be ruled out in some strata, symmetry was supported throughout. Visual inspections of the strata-specific kernel density and standard Gaussian curves showed

**Table 2. Percentages, means (μ) and standard deviate (SD) for HGS (kgf) of the population with successful aging, per sampled cities.**

| Cities | N (%) | $\mu \pm SD$ | $\Delta\mu$ [a] | $\Delta_{SD}$ [b] |
|---|---|---|---|---|
| Barueri | 216 (7.2) | 24.12 ± 8.91 | −1.20 | −0.35 |
| Belém | 275 (9.2) | 23.80 ± 8.04 | −1.52 | −1.22 |
| Belo Horizonte | 254 (8.5) | 25.79 ± 9.21 | 0.47 | −0.05 |
| Campina Grande | 137 (4.6) | 24.70 ± 8.19 | −0.62 | −1.07 |
| Campinas | 393 (13.1) | 27.63 ± 9.51 | 2.31 | 0.25 |
| Cuiabá | 177 (5.9) | 23.75 ± 9.22 | −1.57 | −0.04 |
| Ermelindo | 159 (5.3) | 27.07 ± 8.74 | 1.75 | −0.52 |
| Fortaleza | 91 (3.0) | 24.30 ± 9.99 | −1.02 | 0.73 |
| Ivoti | 94 (3.1) | 28.42 ± 8.31 | 3.09 | −0.95 |
| Juiz de Fora | 129 (4.3) | 18.25 ± 9.22 | −7.07 | −0.04 |
| Parnaíba | 135 (4.5) | 28.56 ± 9.28 | 3.24 | 0.02 |
| Poços de Caldas | 185 (6.2) | 30.32 ±10.33 | 5.00 | 1.07 |
| Recife | 85 (2.8) | 23.35 ± 8.29 | −1.97 | −0.97 |
| Ribeirão Preto | 208 (6.9) | 23.98 ± 8.90 | −1.34 | −0.36 |
| Santa Cruz | 140 (4.7) | 24.48 ± 8.48 | −0.84 | −0.78 |
| Rio de Janeiro | 321 (10.7) | 24.24 ± 8.07 | −1.08 | −1.18 |
| Total | 2999 (100.0) | 25.32 ± 9.26 | — | — |

[a] Distance from grand-mean.

[b] Distance from Standard Deviation (SD) overall (aggregate) average.

remarkable overlapping across most of the value ranges. Since the main analyzes also involved robust procedures (as conveyed before), we thus decided to proceed without any transformation.

The homoscedasticity test suggested rejection in four out of six strata. As in the sequence outlined above, the six strata-specific p-values for testing a constant SD were 0.1068, 0.0203, 0.0111, 0.6339, 0.0017 and 0.0377, respectively. As mentioned in the Methods section, heteroscedasticity led us to indirectly model the varying SD using the coefficient of variation (CV) instead.

Table 3 provides the mean and dispersion regression coefficients, according to the best-fitting FPs. In all strata and age A, these were $\beta_{0(m)}+\beta_{1(m)}\times A^1$ for the mean and $\beta_{0(cv)}+\beta_{1(cv)}\times A^1 = \beta_{0(cv)}$ for the CV because $\beta_{1(cv)} = 0$. Since SD = CV × mean and simplifying, the model-predicted HGS is given by $g = (\beta_{0(m)}+\beta_{1(m)}A^1)(z\beta_{0(cv)}+1)$, where z is the standard score (e.g., values of −1.960, −1.645, 0 or 1.960 relating to the 2.5th, 5th, 50th or 97.5th centiles, respectively). The centile c corresponding to a particular subject A years old with an HGS of g is given by $c = \Phi(-(\beta_{1(m)}A + \beta_{0(m)} - g)/(\beta_{1(m)}\beta_{0(cv)}A + \beta_{0(m)}\beta_{0(cv)}))$, where subscript (m) refers to the mean, (cv) for CV, and (0 or 1) respectively indicate the intercept and slope coefficients in the simple adjusted polynomial. Φ stands for the standard normal distribution function.

So, consistently across strata, the best-fitting models for the means turned out to be first order FP of power 1 (i.e., simple linear regressions), while the FP models for the CV suggested age invariance as indicated by the zero-valued regression slope coefficients. Note, however, that much in line with the detected heteroscedasticity, the SD vary with age as they not only depend on the invariant CV coefficients but also on the means, which are here age dependent as shown by the negative regression coefficients. Table 3 also conveys the sex and height differentials regarding model-based initial values of HGS (at 65 years of age) and their decline with aging.

**Table 3. Means (μ) and SD of hand grip strength (kgf) regarding population reference sample (population with successful aging), per sampled cities.**

| Strata / estimate | | Regression coefficients (equations) | Estimated HGS (kgf) | | Δ [a] |
|---|---|---|---|---|---|
| | | | at age 65 | at age 90 | |
| Male, >1.7 m | | | | | |
| | Mean [b] | $69.29206 - 0.4679578^1 \times A$ | 38.87 [d] | 27.18 | 11.69 |
| | CV [c] | 0.2127911 | | | |
| Male, >1.6–1.7 m | | | | | |
| | Mean | $63.51915 - 0.4171536^1 \times A$ | 36.40 | 25.97 | 10.43 |
| | CV | 0.2150946 | | | |
| Male, ≤1.6 m | | | | | |
| | Mean | $53.20894 - 0.3307964^1 \times A$ | 31.71 | 23.43 | 8.27 |
| | CV | 0.2374751 | | | |
| Females, >1.6 m | | | | | |
| | Mean | $46.55271 - 0.3312745^1 \times A$ | 25.02 | 16.73 | 8.28 |
| | CV | 0.2309019 | | | |
| Female, >1.5–1.6 m | | | | | |
| | Mean | $38.45670 - 0.2517654^1 \times A$ | 22.09 | 15.80 | 6.29 |
| | CV | 0.2418421 | | | |
| Female, ≤1.5 m | | | | | |
| | Mean | $28.51931 - 0.139393^1 \times A$ | 19.45 | 15.97 | 3.48 |
| | CV | 0.2372299 | | | |

[a] Difference between estimated HGS (*kgf*) at age 90 and age 65.

[b] Best-fitting factional polynomial for the mean: $\beta_{0(m)} + \beta_{1(m)} \times A^1$, where A = age. Same solution found for all strata.

[c] Best-fitting factional polynomial for the Coefficient of Variation: $\beta_{0(cv)} + \beta_{1(cv)} \times A^1 = \beta_{0(cv)}$ because $\beta_{1(cv)} = 0$, where A = age. Same solution found for all strata.

All models proved to be adequate. Standardized residuals (z-scores) showed age-independent standard normal distributions, with means very close to zero and SD of 1. The z-scores by age scattergrams and the other diagnostic plots did not spot any departure from normality either. S2 Fig illustrates the visual inspection of plots concerning males, >1.6–1.7m. A very similar pattern was found in all strata (not shown). There was also a noteworthy congruence between the observed and expected percentages falling within the –1 to +1 and –2 to +2 SD ranges. Recalling the Gaussian expected percentages are, respectively, 68.3% and 95.5%, the observed percentages were: (a) male, >1.7: 68.9 / 94.2; (b) male, >1.6–1.7: 69.8 / 94.1; (c) male, ≤1.6: 69.6 / 93.2; (d) females, >1.6: 69.3 / 92.8; (e) female, >1.5–1.6: 68.9 / 93.8; and (f) female, ≤1.5: 71.2 / 92.4.

Both mean and CV equations in tandem enable calculating any desired centile. Thus, for instance, a 65-year-old male, 1.72 meters tall, standing at the median HGS distribution for his age group would have an estimated value of $(69.29206 - 0.46795781 \times 65) \times (0 \times 0.2127911 + 1)$ = 38.87 (as shown in Table 3, 4th column). Generalizing, one may obtain the desired curves, as depicted in Fig 1. Implied by the model specification and ensuing estimated equations, HGS progresses downwards in straight lines, but the intercentile range narrows with age. The sex-height strata, age-specific projected HGS for a wide array of centiles may be found in the S1–S6 Tables. The same applies to plain charts (without scatter dots) to be used in population follow ups if desired (S3–S8 Figs).

Confronting the expected (under the SA assumption) percentages of individuals falling within the –1 and +1 SD and –2 and +2 SD boundaries with the effectively observed in those with UA (validation sample) revealed a consistent pattern across strata. In contrast to a population aging successfully (68.3% and 95.5%, respectively), the observed percentage were always

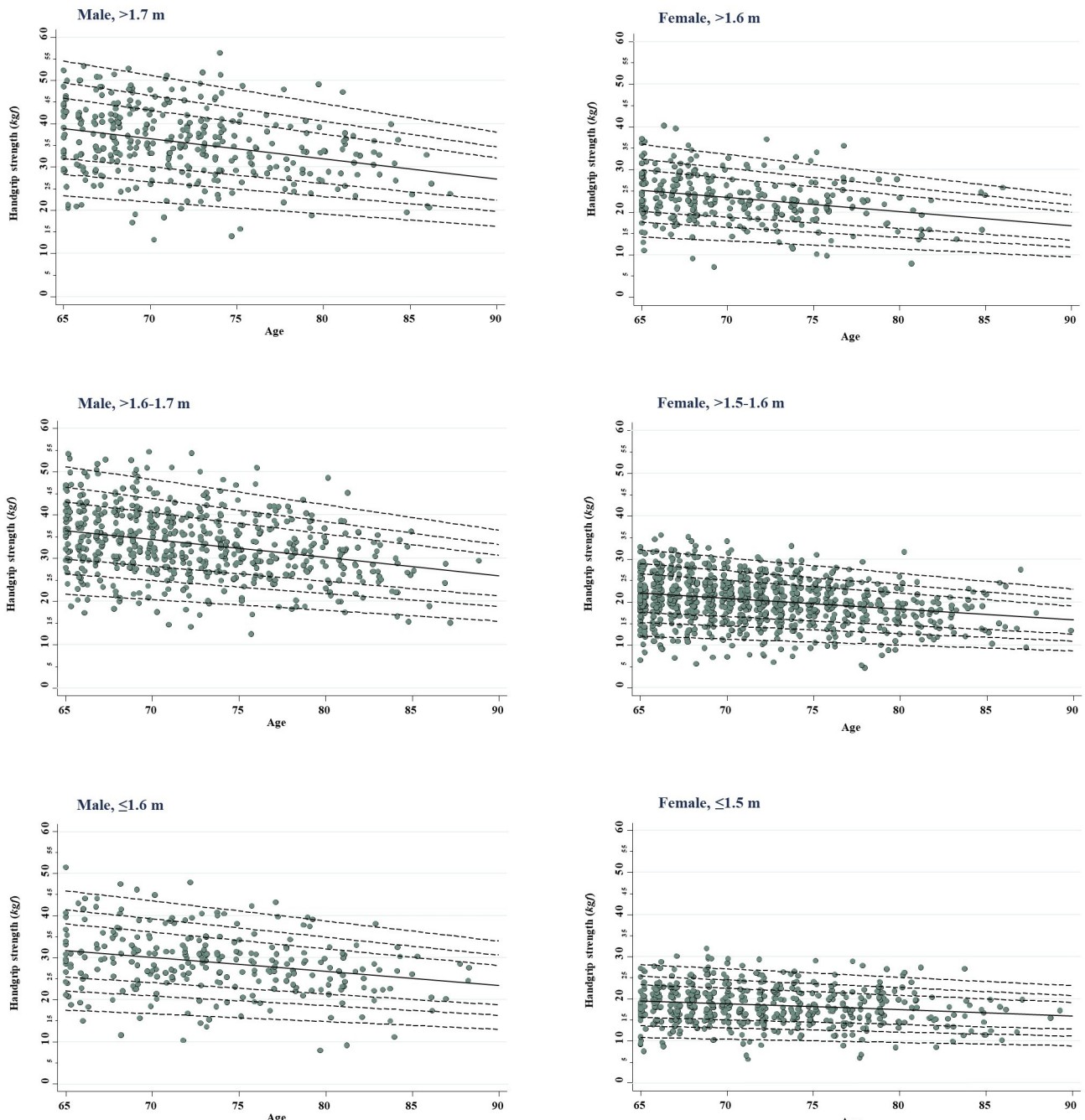

**Fig 1. Centile chart for hand grip strength (*kgf*) regarding population reference sample (population with successful aging), per sex and height strata.**

lower: (a) male, >1.7: 59.7 / 88.2; (b) male, >1.6–1.7: 55.2 / 86.9; (c) male, ≤1.6: 56.9 / 91.5; (d) females, >1.6: 60.5 / 84.1; (e) female, >1.5–1.6: 61.9 / 90.1; and (f) female, ≤1.5: 61.1 / 89.5.

We also foresaw finding fewer debilitated older persons above the upper centiles (80[th], 90[th] and 97[th]), while more than expected below the lower cut-offs (20[th], 10[th] and 3[rd]). As shown in Table 4, this occurred in most situations. For instance, 11.8% of men >1.7 meters tall fell below the 3[rd] centile, while only 3% would be expected in a 'healthy' population. Similarly, an

**Table 4. Population with unsuccessful aging: Contrasting percentage expected and observed by centile threshold (below or above).**

| Centile [a] (%) | | Observed [b] (%) | | | | | |
|---|---|---|---|---|---|---|---|
| | | Male height | | | Female height | | |
| | | >1.7 m | >1.6–1.7 m | ≤1.6 m | >1.6 m | >1.5–1.6 m | ≤1.5 m |
| 97 | (3) | 4.2 (1.7–9.8) | 4.6 (2.8–7.5) | 1.1 (0.2–4.1) | 3.8 (1.7–8.3) | 3.9 (2.8–5.4) | 2.9 (1.9–4.4) |
| 90 | (10) | 5.0 (2.3–10.9) | 10.7 (7.8–14.5) | 5.9 (3.2–10.3) | 7.6 (4.4–13.0) | 9.4 (7.6–11.6) | 7.2 (5.6–9.4) |
| 80 | (20) | 14.3 (9.0–21.9) | 15.8 (12.3–20.2) | 16.0 (11.4–22.0) | 13.4 (8.9–19.7) | 16.5 (14.1–19.2) | 13.8 (11.5–15.5) |
| 20 | (20) | 34.5 (26.4–43.5) | 35.4 (30.4–40.7) | 35.1 (28.6–42.2) | 34.4 (27.3–42.2) | 28.4 (25.5–31.6) | 32.3 (29.0–35.8) |
| 10 | (10) | 23.5 (16.7–32.1) | 25.0 (20.6–30.0) | 22.9 (17.4–29.5) | 21.0 (15.3–28.1) | 18.6 (16.1–21.5) | 18.2 (15.6–21.2) |
| 3 | (3) | 11.8 (7.1–19.0) | 10.1 (7.2–13.8) | 8.5 (5.3–13.5) | 13.4 (8.9–19.7) | 9.3 (07.56–11.5) | 9.4 (7.5–11.8) |

[a] In brackets, percentage expected in a successfully ageing population. % <u>above</u> the 80th, 90th and 97th; and <u>below</u> the 20th, 10th and 3rd centile.

[b] Percentage effectively observed in the unsuccessfully ageing study population. In brackets: 95% Confidence intervals.

excess 14.4% of women ≤1.5 meters tall were found below the 20th centile. This pattern is most apparent in plotting the unsuccessfully ageing older people on the modeled standard curves (S9 Fig). As compared to the reference pattern shown in Fig 1, the scatter plots are consistently shifted downwards in all six strata.

## Discussion

This study proposed HGS values to be used as a reference for normality of the muscular system, both as an indicator for focal, stand-alone assessments, and to evaluate trajectories of older people along their contacts with health services. The values are presented as equations to calculate individual centiles based on measured HGS and in respective centile charts/tables for consultation, stratified by sex, height strata and a refined age discretization. The present study covers subtleties not yet addressed in previous investigations [2, 4, 5, 12, 13].

This study advances the literature by two points. One relates to the NRV/chart development process in general. Although there are previous experiences in this regard, using a stricter definition for the reference population is a differential. Another headway is more 'local' in nature as it seeks to address ethnic, phenotypic, or cultural differences. Although two previous research focused NVR for HGS in Brazil, this is the first study offering reference for broader use, possibly across the country, due to the broad scope of the sampling sites.

In the introduction, we discerned literature in two camps meriting illustration considering the NVR/charts generated here. An example to support the criticism of the use of NRV stratified by sex and only a few age groups (e.g., at ten years intervals) would be appropriate. For example, how would a 71-year-old woman, 1.46 meters tall and measuring 12.6 kgf be thus evaluated and classified? Using this broad age stratification (65–75 years interval) and without any height differentiation, she would, by definition, have the same centile rank as another 66-year-old female 1.82m tall. This is quite different from applying charts as those belonging to the second camp allude to in the Introduction, including the present ones. The equations proposed in our study would place the first woman at 8.6th centile and the second at 1.6 centile, thus possibly classifying them in two distinct risk groups.

Reference values also allow observing the behavior of the HGS variable on a continuum of age by strata. In this perspective, beyond a single measurement (as described above), the chart may be used for decision making as a prospective tool to follow up older persons over time. Contrasting two practical situations illustrates this usage and how different decisions may come about. The first concerns the same 71-year-old female mentioned above, measured for the first time in a primary care unit. Her plotting on the chart (8.6th centile) would suggest

subjecting her to a closer scrutiny, assuming for the sake of argument, that the tenth centile meant delimiting some action to take. Say she was placed under surveillance, having her HGS measured repeatedly over some time without a remarkable change detected (e.g., an HGS measure of 12.2 kgf a year later, which again corresponds to 8.6th centile). Focusing on the trajectory, this picture would lead the health professional to ascertain a normal situation according to the normative standards of aging.

In a second hypothetical situation, a 1.82-meters-68-year-old man shows an HGS of 59.8 kgf on his first contact at a health facility. The corresponding 96th centile would put him in a 'comfort zone' without requiring further scrutiny. Returning to service for reevaluation sometime later, he shows an HGS drop to 37.7 kgf and thus placed on the 53.5th centile. This declining HGS trajectory would suggest a progressive impairment of muscle function and, therefore, in urgent need for further diagnostic scrutiny and therapeutic intervention. Yet, this patient would not have ever been considered at risk if assessed in a cross-sectional evaluation perspective (again, given the tenth centile as a boundary for action taking).

In short, despite an early warning, the trajectory outlined in condition 1 did not involve failure. The first impression of functional success shown in condition 2, however, would not hold in the long run. While none of the identified centiles came any close to warning levels if regarded as a stand-alone measure, a failing trajectory would be apparent. These markedly different scenarios stress how isolated approaches and fixed cut-off points may be insufficient to capture truly dynamic processes. Better insights and thus enhance decision-making processes may be better gauged through trajectory patterns using reference charts and sequential NRV. Clearly, ignoring height and coarsely stratifying the age strata would have further undermined this assessment.

The results of the present study should be seen in light of its strengths and limitations. Besides the advancement posited at the beginning of this section, another positive side concerns the underlying data. To ensure quality data, the Fibra-BR study used a single protocol, involving centralized training in each city, employing standardized equipment and consistent measurement procedures, carefully supervised during fieldwork. Another positive aspect relates to the use for validation of a well-defined population regarded as aging unsuccessfully.

The cross-sectional design is a potential limitation of the study. Although one may assume that a single HGS measure at age $A$ of a subject falling on a given centile relates to a measure of another subject at age $A+t$ falling on the same (or nearby) centile, a longitudinal study design involving repeated HGS measures would have been better suited in providing NRV. Future longitudinal research may help to test the current NRVs and respective charts or, perhaps, provide improvements if required. Besides replicating the analysis on other SA and AU populations, it would also be auspicious to assess the external validity of the NRV in relation to adverse health outcomes, *viz.*, quality of life, falls, dependence, cognitive disorders, multi-morbidities, mortality.

Underscoring the motivations expressed in the Introduction, the NRV for HGS suggested here could help in assisting professionals in the decision-making process concerning the functional status of individuals seeking attention in public and private healthcare services. Managers could also benefit from the charts when establishing priorities in their assignment area. For example, including HGS in repeated surveys may help in situational diagnostics; plotting participants on charts would facilitate planning actions for different risk groups, given the human and financial resources at hand.

Despite some unresolved issues, our NRV/charts may already be endorsed for use in routine services, while further tested and gradually improved in the process. They would aid professionals caring for older people, not only to identify those at risk and eligible for immediate provisions, but also in planning prevention and rehabilitation measures. We believe these standards add to the previous ones, broadening the set of references used to classify older people.

This is especially valid for Brazil, although much could be gained from expanding this research program to other socio-cultural milieus.

## Supporting information

**S1 Checklist. STROBE statement—checklist of items that should be included in reports of *cross-sectional studies.***
(PDF)

**S1 Fig. Distances from the grand mean and standard deviations, per sampled cities.**
(TIF)

**S2 Fig. The z-scores by age scattergrams and the other diagnostic plots concerning the male 1.6–1.7 stratum.**
(TIF)

**S3 Fig. Centile chart for hand grip strength (kgf) for males >1.7 meters.**
(TIF)

**S4 Fig. Centile chart for hand grip strength (kgf) for males >1.6–1.7 meters.**
(TIF)

**S5 Fig. Centile chart for hand grip strength (kgf) for males ≤1.6 meters.**
(TIF)

**S6 Fig. Centile chart for hand grip strength (kgf) for females >1.6 meters.**
(TIF)

**S7 Fig. Centile chart for hand grip strength (kgf) for females >1.5–1.6 meters.**
(TIF)

**S8 Fig. Centile chart for hand grip strength (kgf) for females ≤ 1.5 meters.**
(TIF)

**S9 Fig. Centile curves for hand grip strength (*kgf*) regarding population validation sample (population with unsuccessfully ageing), per sampled cities.**
(TIF)

**S1 Table. Hand grip strength (kgf) projected for male >1.7 meters for a wide array of centiles.**
(DOCX)

**S2 Table. Hand grip strength (*kgf*) projected for male from >1.6 to 1.7 meters for a wide array of centiles.**
(DOCX)

**S3 Table. Hand grip strength (*kgf*) projected for male ≤1.6 meters for a wide array of centiles.**
(DOCX)

**S4 Table. Hand grip strength (kgf) projected for female >1.6 meters for a wide array of centiles.**
(DOCX)

**S5 Table. Hand grip strength (*kgf*) projected for female from >1.5 to 1.6 meters for a wide array of centiles.**
(DOCX)

**S6 Table. Hand grip strength (*kgf*) projected for female ≤1.5 meters for a wide array of centiles.**
(DOCX)

## Acknowledgments

The authors would like to acknowledge the participants of the Frailty in Brazilian Older People Study (Fibra-BR).

## Author Contributions

**Conceptualization:** Roberto Alves Lourenço, Janaína Santos Nascimento, Anita Liberalesso Neri, Lygia Paccini Lustosa, Eduardo Ferriolli.

**Data curation:** Roberto Alves Lourenço, Virgílio Garcia Moreira.

**Formal analysis:** Michael Eduardo Reichenheim, Roberto Alves Lourenço, Janaína Santos Nascimento, Virgílio Garcia Moreira.

**Funding acquisition:** Roberto Alves Lourenço, Anita Liberalesso Neri, Lygia Paccini Lustosa, Eduardo Ferriolli.

**Investigation:** Michael Eduardo Reichenheim, Roberto Alves Lourenço, Janaína Santos Nascimento, Rodrigo Martins Ribeiro, Eduardo Ferriolli.

**Methodology:** Michael Eduardo Reichenheim, Roberto Alves Lourenço, Janaína Santos Nascimento, Virgílio Garcia Moreira, Rodrigo Martins Ribeiro.

**Project administration:** Roberto Alves Lourenço, Janaína Santos Nascimento.

**Resources:** Roberto Alves Lourenço, Janaína Santos Nascimento, Virgílio Garcia Moreira, Anita Liberalesso Neri, Lygia Paccini Lustosa.

**Software:** Michael Eduardo Reichenheim.

**Supervision:** Michael Eduardo Reichenheim, Roberto Alves Lourenço, Janaína Santos Nascimento, Virgílio Garcia Moreira.

**Validation:** Michael Eduardo Reichenheim, Roberto Alves Lourenço, Janaína Santos Nascimento.

**Visualization:** Michael Eduardo Reichenheim, Roberto Alves Lourenço.

**Writing – original draft:** Michael Eduardo Reichenheim, Roberto Alves Lourenço, Janaína Santos Nascimento, Virgílio Garcia Moreira.

**Writing – review & editing:** Michael Eduardo Reichenheim, Roberto Alves Lourenço, Janaína Santos Nascimento, Virgílio Garcia Moreira, Anita Liberalesso Neri, Rodrigo Martins Ribeiro, Lygia Paccini Lustosa, Eduardo Ferriolli.

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
