## [Decision Letter · Decision Letter 0]

4 Feb 2021

PONE-D-20-12109

Normative reference values of handgrip strength for Brazilian older people aged 65 to 90 years: evidence from the multicenter Fibra-BR study

PLOS ONE

Dear Dr. Moreira,

Thank you for submitting your manuscript to PLOS ONE. After careful consideration, we feel that it has merit but does not fully meet PLOS ONE’s publication criteria as it currently stands. Therefore, we invite you to submit a revised version of the manuscript that addresses the points raised during the review process.

Finally, we have the observations of two reviewers. I appreciate your patience in waiting for the opinions. Both reviewers agree that some clarifications are necessary, I suggest that you make them to improve the manuscript.

We look forward to receiving your revised manuscript.

Kind regards,

Martha Asuncion Sánchez-Rodríguez, PhD

Academic Editor

PLOS ONE

Journal Requirements:

2. Please include in your methods section, the full names of the four ethics committees that reviewed and approved this study.

3.We note that the grant information you provided in the ‘Funding Information’ and ‘Financial Disclosure’ sections do not match.

4.Thank you for stating the following in the Competing Interests section:

"Study concept and design – ME Reichenheim, JS Nascimento VG Moreira and RA Lourenço; Acquisition of data – VG Moreira, E Ferrioli, A Liberalesso, L Paccini and RA Lourenço; Analysis and interpretation of data –, ME Reichenheim, JS Nascimento and RA Lourenço; Drafting of the manuscript – ME Reichenheim, JS Nascimento, RM Ribeiro and RA Lourenço; Critical revision of the manuscript for important intellectual content – ME Reichenheim, JS Nascimento, VG Moreira and RA Lourenço. All authors have read and approved the final manuscript."

5. We note you have included a table to which you do not refer in the text of your manuscript. Please ensure that you refer to Table 1, 2 and 4 in your text; if accepted, production will need this reference to link the reader to the Table.

6. Please upload a copy of Supporting Information Table S1, S2, S3, S4 ,S5 and S6 which you refer to in your text on page 27.

Additional Editor Comments:

Finally, we have the observations of two reviewers. I appreciate your patience in waiting for the opinions. Both reviewers agree that some clarifications are necessary, I suggest that you make them to improve the manuscript.

Reviewers' comments:

Reviewer's Responses to Questions

**Comments to the Author**

1. Is the manuscript technically sound, and do the data support the conclusions?

Reviewer #1: Yes

Reviewer #2: Yes

2. Has the statistical analysis been performed appropriately and rigorously? 

Reviewer #1: I Don't Know

Reviewer #2: Yes

3. Have the authors made all data underlying the findings in their manuscript fully available?

Reviewer #1: Yes

Reviewer #2: No

4. Is the manuscript presented in an intelligible fashion and written in standard English?

Reviewer #1: Yes

Reviewer #2: Yes

5. Review Comments to the Author

Reviewer #1: This study has provided NRV of HGS of older persons in a very innovative manner and its results are highly relevant for prevention and rehabilitation strategies.

I only have a few minor revision suggestions :

1. Change the word 'subjects' to 'participants'.

2. line 301 , change the word 'elderly' to older people/persons.

3. Pls add if HGS measurement followed standard procedures recommended by the American Society of Hand Therapists (ASHT) .

4. Please add a section in discussion of if the NRV HGS results were comparable to other previous results.

Reviewer #2: In line 221, in Table 1, there are inconsistencies that need to be clarified:

1) The numbers in some of the characteristics of the total sample are different, it is important to specify why at the footer.

(2) The numbers of the sum of the reference sample and the validation sample do not correspond in all the cases with 3128, and 2447, respectively.

3) The percentages of the total sample are calculated per column, while the percentages in the reference sample and in the validation sample, are not clear how they were calculated.

I could not find original data.

6. PLOS authors have the option to publish the peer review history of their article (what does this mean?). If published, this will include your full peer review and any attached files.

Reviewer #1: **Yes: **Devinder Kaur Ajit Singh

Reviewer #2: No

---

## [Author Response · Author response to Decision Letter 0]

8 Apr 2021

Comments from the editor

a. All adjustments were made.

2. Please include in your methods section, the full names of the four ethics committees that reviewed and approved this study. 

a. Information added (lines 122-125, page 7.

a. This information has been corrected in the cover letter as well as in the article. The correct grant numbers have been entered in the “funding information” section (line 387).

4. We note you have included a table to which you do not refer in the text of your manuscript. Please ensure that you refer to Table 1, 2 and 4 in your text; if accepted, production will need this reference to link the reader to the Table. 

a. All four tables were cited in the first version of the manuscript (lines in 226, 227,250 and 295). In the current version, allusions in the text to Tables 1 to 4 are now in line.

5. Please upload a copy of Supporting Information Table S1, S2, S3, S4, S5 and S6 which you refer to in your text on page 27. 

a. This information was entered in the specific field.

6. Sample validation 

a. Adjustment was done. The sample validation was compounded by older people with low cognitive performance

Comments from reviewers

Reviewer#1:

1. Change the word 'subjects' to participants. 

a. Modified.

2. Line 301 change the word 'elderly' to older people/persons.

a. Modified. 

3. Pls add if HGS measurement followed standard procedures recommended by the American Society of Hand Therapists (ASHT).

a. Information added (line 161, page 8).

4. Please add a section in discussion of if the NRV HGS results were comparable to other previous results. 

a. Information added (line 312, page 17 to line 318, page 18).

Reviewer #2: In line 221, in Table 1, there are inconsistencies that need to be clarified:

1. The numbers in some of the characteristics of the total sample are different, it is important to specify why at the footer. The numbers of the sum of the reference sample and the validation sample do not correspond in all the cases with 3128, and 2447, respectively.

a. We point this out in the footer of Table 1. Please note that the variables of interest (sex, age, height and HGS) do not hold missing data.

3. The percentages of the total sample are calculated per column, while the percentages in the reference sample and in the validation sample, are not clear how they were calculated.

a. All adjustments were made.

---

## [Editor Report · Decision Letter 1]

19 Apr 2021

Normative reference values of handgrip strength for Brazilian older people aged 65 to 90 years: evidence from the multicenter Fibra-BR study

PONE-D-20-12109R1

Dear Dr. Moreira,

We’re pleased to inform you that your manuscript has been judged scientifically suitable for publication and will be formally accepted for publication once it meets all outstanding technical requirements.

Kind regards,

Martha Asuncion Sánchez-Rodríguez, PhD

Academic Editor

PLOS ONE
---

## [Editor Report · Acceptance letter]

23 Apr 2021

PONE-D-20-12109R1 

Normative reference values of handgrip strength for Brazilian older people aged 65 to 90 years: evidence from the multicenter Fibra‑BR study 

Dear Dr. Moreira:

I'm pleased to inform you that your manuscript has been deemed suitable for publication in PLOS ONE. Congratulations! Your manuscript is now with our production department. 

Kind regards, 

on behalf of

Dr. Martha Asuncion Sánchez-Rodríguez 

Academic Editor

PLOS ONE